# Neoadjuvant Chemohormonal Therapy before Radical Prostatectomy for Japanese Patients with High-Risk Localized Prostate Cancer

**DOI:** 10.3390/medsci9020024

**Published:** 2021-04-09

**Authors:** Takeshi Sasaki, Kouhei Nishikawa, Manabu Kato, Satoru Masui, Yuko Yoshio, Yoshiki Sugimura, Takahiro Inoue

**Affiliations:** 1Department of Nephro-Urologic Surgery and Andrology, Mie University Graduate School of Medicine, Mie 514-8507, Japan; t-sasaki@med.mie-u.ac.jp (T.S.); katouuro@clin.medic.mie-u.ac.jp (M.K.); s-masui@med.mie-u.ac.jp (S.M.); y-yoshio@clin.medic.mie-u.ac.jp (Y.Y.); tinoue28@med.mie-u.ac.jp (T.I.); 2Urology and Prostate Center, Murase Hospital, Mie 513-0801, Japan; sugimura@murase.or.jp

**Keywords:** prostate cancer, neoadjuvant chemohormonal therapy, radical prostatectomy

## Abstract

Background: Radical prostatectomy (RP) is the standard treatment in patients with high-risk prostate cancer (PCa). However, there is a high rate of recurrence, and new approaches are required to improve surgical efficacy. Here, we evaluated the feasibility and safety of neoadjuvant chemohormonal therapy (NCHT) before RP for Japanese patients with high-risk localized prostate cancer (PCa). Methods: From February 2009 to April 2016, 21 high-risk patients were enrolled in this prospective study. Patients were treated with docetaxel (70 mg/m^2^) every four weeks for three cycles and luteinizing hormone-releasing hormone agonist. Patients with grade 3–4 toxicities had 25% dose reductions for the following course. Results: Median follow-up was 88.6 months. The dose of docetaxel was reduced in 13 patients. The estimated five-year biochemical progression-free survival (bPFS) rate was 57.1%. National Comprehensive Cancer Network criteria (high-risk, but not very high-risk (nVHR) versus VHR) was associated with bPFS (*p* = 0.03). Five-year bPFS rates in the nVHR and VHR groups were 76.9% and 25.0%, respectively. There was a significant difference in bPFS between the nVHR and VHR groups (*p* = 0.023) by Kaplan–Meier analysis. Conclusions: Although our study included a small number of cases, at least in our exploration, NCHT was safe and feasible. However, more extensive treatment modalities are needed to improve outcomes, especially in VHR patients.

## 1. Introduction

Although radical prostatectomy (RP) against high-risk prostate cancer (PCa) is considered a standard treatment, recurrence frequently occurs, leading to cancer-related mortality [1,2]. Patients with high-risk PCa treated with RP alone have a 55.2% risk of five-year biochemical progression and a 14.8% risk of 10-year cancer-specific mortality [2]. Among men with high-risk PCa, recent evidence has shown that very high-risk (VHR) patients stratified by the National Comprehensive Cancer Network (NCCN) criteria appear to have distinctly worse oncologic outcomes after RP [3].

To improve surgical efficacy, neoadjuvant trials have been performed before RP in patients with high-risk PCa. According to multiple randomized trials, neoadjuvant hormone therapy (NHT) with RP does not result in significant improvement compared with prostatectomy alone [4]. Recently, a phase II trial using enzalutamide and leuprolide with or without abiraterone and prednisone before RP reported that NHT with potent AR-directed therapies resulted in favorable pathologic responses in a subset of patients with high-risk disease [5]. Future validation of phase III trials will evaluate post-RP outcomes with intense NHT.

Several small clinical studies have reported that neoadjuvant chemohormonal therapy (NCHT) with docetaxel plus androgen deprivation therapy (ADT) is safe and feasible, and promotes favorable pathologic outcomes in high-risk localized PCa [6,7,8,9,10,11,12,13]. The concept of NCHT involves the treatment of both androgen-dependent and -independent PCa cells, both locally and systemically, before RP. The Cancer and Leukemia Group B/Alliance 90,203 trial [14] is a phase III study in which 778 men with high-risk PCa were randomly assigned in a 1:1 fashion based on Gleason grade group or Kattan’s nomogram [15]. The primary endpoint was three-year biochemical progression-free survival (bPFS) after NCHT. Notably, in this study, 47% of patients were not included in the evaluation of the primary endpoint because they had received additional treatment before meeting the primary endpoint. Although the primary endpoint was not statistically significant, there was improvement in overall bPFS, metastasis-free survival, and overall survival (OS) in the NCHT arm [14]. Moreover, a longer duration of follow-up may have yielded more significant results regarding improvement of bPFS and OS in the NCHT arm. The results strongly suggested that NCHT before RP has a substantial potential to improve outcomes for high-risk PCa patients.

Additionally, although a series of studies have been conducted to evaluate the efficacy of NCHT with regard to disease progression and survival, little is known about its relationship to the clinical and pathological characteristics of patients. In this prospective study, we evaluated the feasibility and safety of NCHT before radical prostatectomy for Japanese patients with high-risk localized PCa, and analyzed whether there were differences in responses to NCHT between clinical and pathologic variables for patients.

## 2. Materials and Methods

### 2.1. Study Design and Patients

From February 2009 to April 2016, 21 patients were enrolled in this prospective study at Mie University Hospital. The inclusion criteria were as follows: proven histological diagnosis of PCa; younger than 75 years of age; World Health Organization (WHO) performance status 0 or 1; NCCN high-risk criteria (prostate-specific antigen (PSA) >20 ng/mL or Gleason score ≥8 or cT3a); without lymph node and distant metastasis; and provided informed consent. This study included a small number of patients (21 patients) who provided full informed consent during a long period because NCHT before RP is still controversial as to whether or not NCHT has a potential to improve clinical outcomes.

All patients received NCHT before RP. Our treatment consisted of intravenous administration of docetaxel (70 mg/m^2^) every four weeks for three cycles and luteinizing hormone-releasing hormone (LHRH) agonist (Leuprorelin 11.25 mg). The dose intensity of docetaxel was decided based on a previous report [16]. Any cases exhibiting grade 3–4 toxicity required a 25% dose reduction for the subsequent course. RP was performed by open or robot-assisted surgery (13 and eight patients, respectively). All patients received RP with limited pelvic lymph node dissection (PLND), and extended PLND (ePLND) was performed in only six patients. PSA was serially measured every three months after prostatectomy. Biochemical progression was defined as two consecutive PSA values greater than 0.2 ng/mL according to the guideline of the Japanese Urological Association. Time to events was calculated from the day of surgery. Only one patient did not have a postoperative PSA less than 0.2 ng/mL. Imaging for metastatic disease was left to clinical judgement based on PSA and/or symptoms of recurrent disease. After biochemical progression, the following treatments were performed: salvage radiation therapy, hormonal therapy (antiandrogen treatments, estrogen treatments), and chemotherapy. Castration-resistant PCa was defined as either progressively rising PSA (resulting in two 50% increases over the nadir, with PSA > 2.0 ng/mL), despite a castrate level (< 50 ng/dL) of testosterone according to a previous report [17]. Toxicity was classified according to the National Cancer Institute Common Toxicity Criteria version 2.0. Pathologic changes after chemohormonal therapy were graded using the general rule for clinical and pathologic studies for PCa as grade 0 (viable cells in all areas), grade 1 (nonviable cells in half or less cancer areas), grade 2 (nonviable cells in more than half of the cancer areas), or grade 3 (nonviable cells or no cancer cells) [18]. 

### 2.2. Statistical Analysis

Univariate analysis was performed according to the Cox proportional hazards regression model. We defined cutoff points regarding age, PSA density, maximum tumor diameter, PSA after NCHT, and resected lymph node number by using the median value, and initial PSA, cT stage, grade group, and NCCN criteria (high-risk, but not very high-risk and very high-risk) for univariate analysis. The grade group included grades as GG1 (Grade Group 1) = Gleason score ≤ 6, GG2 = Gleason score 3 + 4 = 7, GG3 = Gleason score 4 + 3 = 7, GG4 = Gleason score 8, and GG5 = Gleason score 9 to 10 according to the 2014 International Society of Urological Pathology Consensus Conference [19]. VHR was defined as cT3b-4, Primary Gleason pattern = 5, multiple high-risk features, >4 cores with Grade Group 4 or 5, with patients stratified by NCCN [20]. Only one variable was found to be significant on univariate analysis (*p* < 0.05), and therefore, multivariate analysis was not performed. Survival probability, which was defined as the time when surgery was performed until the date of progression, was calculated with the Kaplan–Meier method. Results with *p*-values of less than 0.05 were considered statistically significant by log-rank tests. Statistical analyses were performed using SPSS software version 22 (IBM Corporation, Armonk, NY, USA), and results with *p*-values of less than 0.05 were considered statistically significant.

## 3. Results

### 3.1. Patients Characteristics

The clinical characteristics of the patients in this study are shown in Table 1. The median age was 65 years (range: 55–73 years), and the median follow-up was 88.6 months (range: 32–121 months). PSA values decreased in all patients following treatment with NCHT. The dose of docetaxel was reduced in 13 patients (61.9%). The median relative dose intensity was 83.3% (range: 66.6–100%).

### 3.2. Analysis of Biochemical Progression-Free Survival and Overall Survival

Twenty patients (95.3%) were alive at the end of the follow-up, and one patient (4.7%) had metastatic disease and died 109.7 months after the day of surgery. Eleven patients (52.4%) remained alive with no biochemical or clinical progression. Ten patients (47.6%) had biochemical progression. Two patients (9.5%) progressed to castration-resistant PCa (Figure 1). The 10 patients who had biochemical progression had subsequent therapies, including salvage radiotherapy, hormonal therapy, chemotherapy, or a combination of these therapies. The estimated five-year bPFS, distant metastasis-free survival, prostate cancer-specific survival, and overall survival (OS) rates were 57.1%, 89.4%, 100%, and 100%, respectively (Figure 1). Based on the pathologic changes of 21 patients, there were no significant differences in bPFS and OS between non-responders (Grade 0) and responders (Grade 1, 2, 3) (data not shown).

### 3.3. Analysis of Factors Predicting Biochemical Progression-Free Survival

We analyzed bPFS predicting factors in 21 patients with locally advanced PCa who received NCHT before RP. Univariate analysis showed that NCCN criteria (high risk, but not VHR (nVHR) versus VHR) were associated with bPFS (*p* = 0.03; Table 2). 

Eight of 21 patients (38%) were in the VHR group, and Kaplan–Meier analysis showed that there was a significant difference in bPFS between the nVHR and VHR groups (*p* = 0.023; Figure 2). The five-year bPFS rates in patients in the nVHR and VHR groups were 76.9% and 25.0%, respectively (Figure 2). The median times to bPFS in the nVHR and VHR groups were 88.6 and 20.4 months, respectively (*p* = 0.013).

### 3.4. Analysis of Peri-Operative Toxicities during NCHT

During NCHT, neutropenia was the most frequently observed toxicity (Table 3). Grade 3–4 toxicities included neutropenia (85%), febrile neutropenia (10%), and mucositis oral (5%). Low-grade anorexia was observed in seven patients (33%) during NCHT. Postoperative adverse events were obturator neuropathy (Clavien–Dindo classification grade II), deep vein thrombosis (grade II), lymphorrhea (grade IIIa), and ureterovesical stricture (grade IIIa).

Pathological changes were grade 0 in nine patients (43%), grade 1 in five patients (24%), grade 2 in seven patients (33%), and grade 3 in zero patients (0%; Table 1). The resection margin was negative in 16 patients (76%), and extraprostatic extension was found in three patients (14%).

## 4. Discussion

In this study, we evaluated predicting factors of bPFS in 21 patients with locally advanced PCa with NCHT before RP. To the best of our knowledge, this is the first study showing that there was a significant difference in bPFS between the nVHR and VHR groups stratified by NCCN criteria after RP with NCHT. A previous study demonstrated that patients with nVHR treated with RP alone had a 55.9% risk of five-year bPFS, and patients with VHR treated with RP alone had a 31.1% risk of five-year bPFS [3]. In our study, a better five-year bPFS rate (76.9%) was observed in the nVHR group. However, NCHT was not acceptable for improving the five-year bPFS of VHR PCa (25.0%). This observation emphasized that additional treatment modalities are needed to improve outcomes in patients with VHR PCa.

We summarized several lines of clinical evidence showing NCHT with docetaxel plus ADT before RP (Table 4) [6,7,8,9,10,11,12,13,14].

This study is the longest follow-up study ever published of NCHT (as shown in Table 4). Most of these studies enrolled patients with high-risk PCa, including VHR patients. Only one retrospective study focused on patients with VHR PCa [13]. In contrast, Zurita et al. evaluated the efficacy of NCHT in patients with aggressive PCa with lymph node metastasis [11]. None of the studies compared the efficacy of NCHT between the nVHR and VHR groups. In patients with high-risk PCa, including VHR patients, previous studies showed that the five-year bPFS rates ranged from 10% to 60.1% (Table 4). Our study demonstrated favorable clinical outcomes, with a five-year bPFS rate of 57.1% (Figure 1). In patients with VHR PCa, the median time to bPFS was 19 months (Table 4). Our study showed similar clinical outcomes, with a median time to bPFS of 20.1 months in the VHR group. Thus, our data represented a reasonable clinical outcome of NCHT with docetaxel plus ADT before RP. Recently, Eastham et al. demonstrated a large prospective randomized study (RP alone versus RP with NCHT) [14]. Although the primary end point, three-year bPFS, was not met, secondary end points (overall bPFS, metastasis-free survival, and overall survival) were positive.

In our analysis of bPFS predicting factors (Table 3), only NCCN criteria (nVHR versus VHR) was significant in univariate analysis (hazard ratio: 4.07, *p* = 0.03). Although Grade Group (4, 5 versus 1, 2, 3; hazard ratio: 5.69, *p* = 0.10), resected lymph node number (≥10, <10; hazard ratio: 1.68, *p* = 0.42), and largest cross-section tumor diameter (≥20 mm, <20 mm; hazard ratio: 3.77, *p* = 0.059) were not statistically significant (Table 3), there is a strong possibility that the statistically non-significant results were caused by the small number of patients. We believe that these are noteworthy variables. Pathologic change of PCa after NCHT was not a significant factor of bPFS and OS in our study. Although previous report also demonstrated pathologic change of PCa after NCHT did not affect bPFS [12], further investigation of pathologic features of NCHT species could elucidate molecular target to predict the NCHT resistance [21,22].

Based on our analysis, in order to improve outcomes, we considered that there might be two important factors. First, more aggressive lymph node dissection could lead to better outcomes. A limitation of this study is that we performed ePLND in only six patients (28%). In our study, the median number of resected lymph nodes was 10 (range: 2–28; Table 1). A previous study showed that the number of resected lymph nodes by ePLND was significantly higher than that of limited PLND (median: 6 (range: 2–9), 16 (range: 13–21), respectively) [23]. By ePLND, resection of a greater number of metastatic lymph nodes might lead to improved oncological outcomes. Although ePLND removes many metastatic nodes compared with limited PLND [24], whether ePLND is associated with a survival benefit remains unclear [25]. Thus, the therapeutic value of PLND with NCHT remains unclear. Second, reducing the largest cross-section tumor diameter, i.e., achieving a better pathological response, could lead to better outcomes. In our study, the pathological outcomes were as follows: 0% complete response (CR), 57% partial response, 43% stable disease, and 0% progressive disease, based on the residual tumor volume defined by the general rule [18]. Previous reports have shown that the pathological CR rate is 0–17.31% (Table 4). Compared with previous studies, our pathological CR rate was poor. A possible explanation for this outcome is that our regimen (total docetaxel dose (210 mg/m^2^) and without anti-androgen therapy) was not sufficient to kill cancer cells. The median docetaxel dose in other regimens is 300 mg/m^2^ (range: 180–630 mg/m^2^), and all other NCHT studies combined docetaxel with not only an LHRH agonist, but also anti-androgen and/or estramustine therapy (Table 4). This may be reflected in PSA titers before prostatectomy. Compared with other studies, our median PSA titer after NCHT (1.28 ng/mL) was much higher [12,13].

The secondary endpoint of this study was adverse events. Pre-operative toxicities during NCHT are shown in Table 2. Because docetaxel plus ADT is the standard therapy against castration-resistant PCa, grade 3–4 toxicities, including neutropenia (85%), febrile neutropenia (10%), and mucositis oral (5%), were expected. Around the time of surgery, there were no specific adverse events. Postoperative adverse events included obturator neuropathy (Clavien–Dindo classification grade II), deep vein thrombosis (grade II), lymphorrhea (grade IIIa), and ureterovesical stricture (grade IIIa).

We recognize the limitations of the small sample size of our study. However, we believe that our long-term study demonstrated the efficacy of NCHT before prostatectomy. Additional prospective studies with large cohort sizes and combination with intense NHT or immuno-oncology drugs with extended lymph node dissection will enable identification of the critical roles of neoadjuvant therapy before RP in prolonging PFS and OS.

## 5. Conclusions

In conclusion, NCHT was safe and feasible. A better bPFS rate was observed in the nVHR group. However, NCHT was not acceptable to improve bPFS in patients with VHR PCa. Additional treatment modalities are needed to improve outcomes in patients with VHR PCa.

## Figures and Tables

**Figure 1 medsci-09-00024-f001:**
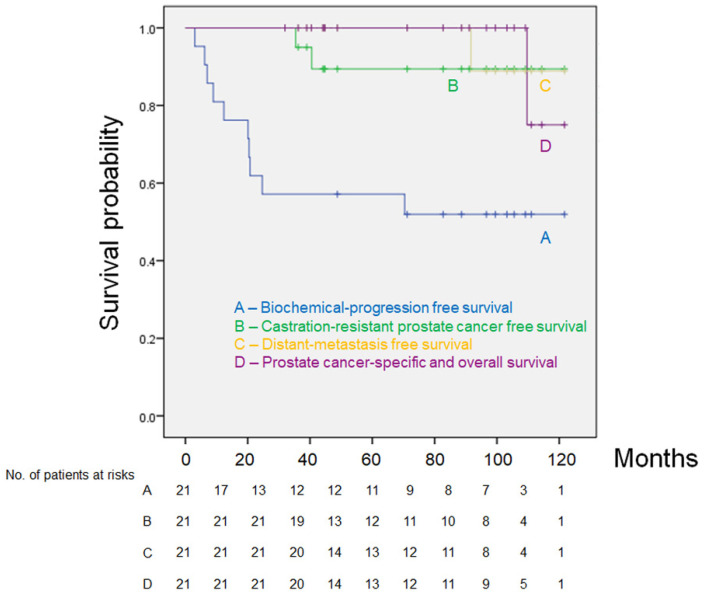
Kaplan–Meier estimates of biochemical progression-free survival (**A**), castration-resistant prostate cancer-free survival (**B**), distant metastasis-free survival (**C**), prostate cancer-specific survival, and overall survival (**D**).

**Figure 2 medsci-09-00024-f002:**
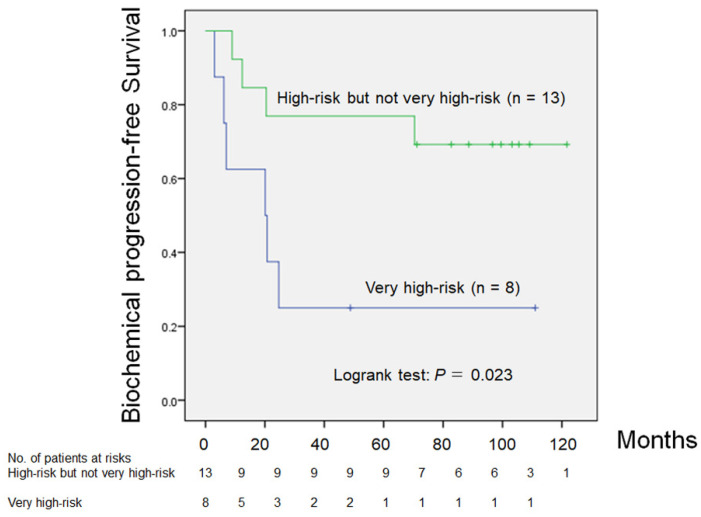
Comparison of biochemical progression-free survival between patients with high risk, but not very high risk PCa versus patients with very high risk PCa (*p* = 0.023).

**Table 1 medsci-09-00024-t001:** Patients and pathologic characteristics at prostatectomy.

Number of Patients	21
Median age (years) (range)	65.3 (55–73)
Median initial PSA (ng/mL) (range)	11.4 (4.3–97.0)
Median PSA density (ng/mL/mL) (range)	0.40 (0.14–4.95)
Clinical T stage at diagnosis (%)	
T2b T2c T3a T3b	1 (5) 3 (15) 16 (75) 1 (5)
Median maximum tumor diameter before NCHT by MRI (mm) (range)	20.0 (13.2–37.7)
Grade group (%)	
1 2 3 4 5	1 (5) 4 (19) 1 (5) 8 (37) 7 (34)
NCCN criteria (%)	
High risk, but not very high risk (PSA > 20 or Grade Group 4 or 5 or T3a with no very high-risk features)	13 (62)
Very high-risk (T3b-4 or Primary Gleason pattern 5 or multiple high-risk features or >4 cores with Grade Group 4 or 5)	8 (38)
Median PSA after NCHT (ng/mL) (range)	1.28 (0.08–7.66)
Pathological T stage (%)	
T2a T2b T2c T3a T3b	8 (37) 2 (10) 7 (33) 2 (10) 2 (10)
Pathological N stage (%)	
0 1	20 (95) 1 (5)
Extraprostatic extension (%)	
Negative Positive	18 (85) 3 (15)
Resection margin (%)	
Negative Positive	16 (76) 5 (24)
Median number of resected lymph nodes (range)	10 (2–28)
Pathologic changes after chemohormonal therapy (%)	
Grade 0 Grade 1 Grade 2 Grade 3	9 (43) 5 (24) 7 (33) 0 (0)

Abbreviations: PSA, prostatic specific antigen; NCHT, neoadjuvant chemohormonal therapy; MRI, magnetic resonance imaging; NCCN, National Comprehensive Cancer Network; RP, radical prostatectomy.

**Table 2 medsci-09-00024-t002:** Univariate analysis of factors predicting biochemical progression-free survival in high-risk localized prostate cancer patients with neoadjuvant chemohormonal therapy before radical prostatectomy (*n* = 21).

Factors	Univariate Hazard Ratio [95% CI] (*p*-Value)
Age (<65 vs. ≥65)	1.07 [0.30–3.82] (*p* = 0.91)
Initial PSA (≥20 vs. <20)	0.96 [0.25–3.73] (*p* = 0.95)
PSA density (≥ 0.4 vs. < 0.4)	0.53 [0.15–1.91] (*p* = 0.33)
cT (≥3 vs. <3)	0.80 [0.20–3.16] (*p* = 0.75)
Maximum tumor diameter before NCHT by MRI (≥20 mm vs. <20 mm)	1.41 [0.39–5.06] (*p* = 0.59)
Grade group (4, 5 vs. 1, 2, 3)	5.69 [0.71–45.4] (*p* = 0.10)
NCCN criteria (very high vs. high risk but not very high)	4.07 [1.11–14.9] (*p* = 0.03)
PSA after NCHT (≥1.28 vs. <1.28)	1.30 [0.37–4.62] (*p* = 0.69)
pT (≥3 vs. <3)	1.05 [0.22–4.98] (*p* = 0.95)
Extraprostatic extension (positive vs. negative)	0.53 [0.06–4.23] (*p* = 0.55)
Resection margin (positive vs. negative)	1.47 [0.38–5.72] (*p* = 0.57)
Resected lymph node number (<10 vs. ≥10)	1.68 [0.48–5.81] (*p* = 0.42)
Largest cross-section tumor diameter (≥20 mm vs. <20 mm)	3.77 [0.94–15.0] (*p* = 0.059)

Abbreviations: NCHT, neoadjuvant chemohormonal therapy; NCCN, National Comprehensive Cancer Network.

**Table 3 medsci-09-00024-t003:** Peri-operative toxicities according to grade during neoadjuvant chemohormonal therapy (*n* = 21).

Grade (NCI-CTC)	0	1	2	3	4
Neutropenia	0	1	2	6	12
Febrile neutropenia	19	0	0	2	0
Mucositis oral	20	0	0	1	0
Malaise	18	3	0	0	0
Anorexia	14	7	0	0	0
Diarrhea	18	2	1	0	0
Aspartate aminotransferase increased	17	3	1	0	0
Peripheral sensory neuropathy	19	2	0	0	0

Abbreviations: NCI-CTC, National Cancer Institute’s Common Terminology Criteria.

**Table 4 medsci-09-00024-t004:** Summary of neoadjuvant docetaxel-based chemohormonal studies before radical prostatecotomy.

Study	Patients	Treatment Regimens	Results	Median Follow-Up Time (Months)
Prayer-Galetti et al., 2007 [6]	*n* = 22, ≥PSA 15 ng/mL or ≥Gleason score 8 or ≥cT3	q3 weeks × 4 cycles docetaxel (70 mg/m^2^) plus triptorelin plus estramustine	15% CR, 80% PR, 5% PD. Five-year rate of bPFS was 42%.	53
Chi et al., 2008 [7]	*n* = 72, ≥PSA 20 ng/mL and/or ≥Gleason score 7 or ≥cT3	Six weekly docetaxel (35 mg/m^2^), two weeks off × 3 cycles, plus buserelin plus nilutamide/flutamide/bicalutamide	3% CR. bPFS was 70% at 43 months.	42.7
Sella et al., 2008 [8]	*n* = 22, ≥PSA 20 ng/mL or ≥Gleason score 8 or ≥cT2c	q3 weeks × 4 cycles docetaxel (70 mg/m^2^) plus goserelin plus bicalutamide plus estramustine	0% CR. 50% of bPFS was 30 months.	23.6
Mellado et al., 2009 [9]	*n* = 57, ≥PSA 20 ng/mL or ≥Gleason score 4+3 or ≥cT3	Three weekly docetaxel (36 mg/m^2^), one week off x 3 cycles plus goserelin plus flutamide	6% CR, 6% near CR. 31.6% patients presented PSA relapse.	35
Thalgott et al., 2014 [10]	*n* = 30, BCR > 40% within five years by Kattan’s preoperative nomogram [15]	q3 weeks × 3 cycles docetaxel (75 mg/m^2^) plus buserelin plus bicalutamide	0% CR, pathological down-staging was observed in 48.3%. Five-year rate of bPFS was 10%. In patients defined as therapy responders, five-year rate of bPFS was 40%.	48.6
Zurita et al., 2015 [11]	*n* = 40, Confirmed lymph node metastasis or suspected lymph node metastasis (Gleason score ≥8 plus ≥PSA 25 ng/mL, cT3 with Gleason score ≥7, cT4)	Six weekly docetaxel (35 mg/m^2^), two-week off × 3 cycles plus LHRH agonist plus bicalutamide	8% CR. The median time to treatment failure was 21.6 months.	61
Narita et al., 2019 [12]	*n* = 60, cT3 or PSA ≥ 15 ng/mL or Gleason pattern 5 (primary and/or secondary)	Six weekly docetaxel (30 mg/m^2^) × 1 cycle plus leuproletin/goserelin plus bicalutamide plus estramustine	10% CR. Five-year bPFS was 60.1%.	42.5
Pan et al., 2019 [13]	*n* = 60, cT3a or Primary Gleason pattern 5, or ≥5 cores with Gleason sum 8 to 10, or PSA ≥50 ng/mL or with pelvic metastatic lymph node involvement	q3 weeks × 4–6 cycles of docetaxel (75 mg/m^2^) plus goserelin plus bicalutamide	17.31% CR. Pathological down-staging 61.5%. The median time to biochemical recurrence was 19 months.	12.5
Eastham et al., 2020 [14]	*n* = 778, Grade Group 4 or 5, or Kattan’s preoperative nomogram [15] bPFS < 60%	q3 weeks × 6 cycles docetaxel (75 mg/m^2^) plus buserelin plus LHRH agonist	No difference was seen in three-year bPFS between RP with NCHT and RP alone.	72.1
Present study	*n* = 21, >PSA 20 ng/mL or ≥Gleason score 8 or cT3a	q4 weeks × 3 cycles docetaxel (70 mg/m^2^) plus leuprorelin	0% CR, 57% PR, 43% SD, 0% PD. Five-year rates of bPFS, DMFS, PCSS, and OS were 57.1%, 89.4%, 100%, and 100%, respectively.	88.6

Abbreviations: PSA, prostate-specific antigen; CR, complete response; PR, partial response; SD, stable disease; PD, progression disease; BCR, biochemical recurrence; LHRH, luteinizing hormone-releasing hormone; bPFS, biochemical progression-free survival; RP, radical prostatectomy; NCHT, neoadjuvant chemohormonal therapy; DMFS, distant metastasis-free survival; PCSS, prostate cancer-specific survival; OS, overall survival.

## Data Availability

The data that support the findings of this study are not available.

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
