# Peer review of "Neoadjuvant Chemohormonal Therapy before Radical Prostatectomy for Japanese Patients with High-Risk Localized Prostate Cancer"

_medsci, 2021, doi:10.3390/medsci9020024_

Round 1

Reviewer 1 Report

The manuscript presents a study of neoadjuvant chemotherapy/androgen ablation therapy in high-risk patients undergoing radical prostatectomy. They concluded that patients with high-risk but not those with very high risk according to NCCN critera, profit from this treatment and analyzed factors associated with biochemical disease progression. While the report is of value and interesting in principle, several points should be addressed to improve the quality and clarity of the manuscript.

The major limitation of the study is the small number of enrolled patients (21). The authors address this in the Discussion, however, in my view it should also be mentioned in the Abstract. 

The authors use a lot of abbreviations. Most readers who are not specialist urologists will be unfamiliar with them. Avoiding unfamiliar abbreviations would increase readability and clarity. 

NCCN risk criteria are crucial to the understanding of the study and the conclusions. All definitions and criteria used should be completely given in Material and Methods. Furthermore, NCCN high-risk criteria described in M&M are different from those listed in Table I. The authors should stay to one precise and consistent nomenclature and not confuse with Gleason Score/Gleason Group/Grade Group.  

The definition for castration-resistant prostate cancer should be given in Material and Methods as well. 

Table 1 should be made clearer, layout and style don't seem ok.

Author Response

Answer to the reviewer 1

1)      The major limitation of the study is the small number of enrolled patients (21). The authors address this in the Discussion, however, in my view it should also be mentioned in the Abstract.

We thank the reviewer’s comment. We have added a new sentence “Although our study included a small number of cases, at least in our exploration, ….” in the ABSTRACT of Page 1, line 35-36.

2)      The authors use a lot of abbreviations. Most readers who are not specialist urologists will be unfamiliar with them. Avoiding unfamiliar abbreviations would increase readability and clarity.

We agree with the reviewer. We have minimized the use of abbreviations.

3)      NCCN risk criteria are crucial to the understanding of the study and the conclusions. All definitions and criteria used should be completely given in Material and Methods. Furthermore, NCCN high-risk criteria described in M&M are different from those listed in Table I. The authors should stay to one precise and consistent nomenclature and not confuse with Gleason Score/Gleason Group/Grade Group.

We thank the critical reviewer’ suggestion. We have added the definition for NCCN risk criteria (high risk and very high risk) and Gleason Score/Grade Group in the Material and Methods of Page 3 in line 119-124 and corrected those lists in Table 1.

4)      The definition for castration-resistant prostate cancer should be given in Material and Methods as well.

We have described the definition for castration-resistant prostate cancer in the Material and Methods of Page 3, line 105-108 according to reviewer’s comment.

5)      Table 1 should be made clearer, layout and style don't seem ok.

We have changed the layout of Table 1 according to reviewer’s comment.

Reviewer 2 Report

Title:

The title reflects the aims of the study well.

Abstract:

The abstract reflects their work and the aims.

Introduction:

Introduction is well-written and easy to follow.

Methods:

Overall, the study design is clear and appropriate for the aims of the study.

Results:

The results were clearly presented and could be followed easily in figure and tables.

There are some points that I would suggest.

・In table 1, I suggested that authors correct the number of Gleason group.

・Is there any difference between grade 0 and grade 1-3 in pathological changes? Even if there is no difference, it is better to mention and discuss it.

Discussion:

The discussion addressed the research question and interpreted the results in a plausible way.

There are some points that I would suggest.

・In table 4, the authors reported the Eastham et al treatment regimen was docetaxel plus buserelin plus LHRH agonist. This study is very important for current study. Please clarify.

Conclusions:

The conclusion reflects their work in a plausible way.

Author Response

Answer to the reviewer 2
1)      The results were clearly presented and could be followed easily in figure and tables.

There are some points that I would suggest.

・In table 1, I suggested that authors correct the number of Gleason group.

・Is there any difference between grade 0 and grade 1-3 in pathological changes? Even if there is no difference, it is better to mention and discuss it.

We have corrected the number of Grade group in Table 1.

We thank the critical reviewer’s suggestion. We reanalyzed our data based on pathologic changes. There were no significant differences in biochemical progression-free survival and overall survival between grade 0 and grade 1-3 patients. We have added a new sentence “Based on pathologic changes of 21 patients, there were no significant differences in bPFS and OS between non-responders (Grade 0) and responders (Grade 1, 2, 3) (data not shown).” in the RESULTS of Page 5 in line 151-153.

Beltran et al. (Clin Cancer Res. 2017 Nov 15;23(22):6802-6811) and Matsuda et al. (BMC Cancer. 2020 Apr 15;20(1):302) investigated molecular pathologic features of NCHT species and provided new insight into potential markers of chemohormonal resistance. We have added new sentences “Pathologic change of PCa after NCHT was not a significant factor of bPFS and OS in our study. Although previous report also demonstrated pathologic change of PCa after NCHT did not affect on bPFS [12], further investigation of pathologic features of NCHT species could elucidate molecular target to predict the NCHT resistance [21,22].” in the DISCUSSION of Page 8 in line 229-233.

2)      The discussion addressed the research question and interpreted the results in a plausible way.

There are some points that I would suggest.

・In table 4, the authors reported the Eastham et al treatment regimen was docetaxel plus buserelin plus LHRH agonist. This study is very important for current study. Please clarify.

We thank the critical reviewer’ suggestion. We agree with the reviewer. Prospective neoadjuvant chemothermal therapy before radical prostatectomy were performed by Eastham et al, this study enrolled 778 patients and the primary end points, 3-year biochemical progression free survival, was not met (J Clin Oncol. 2020 Sep 10;38(26):3042-3050). But secondary end points (overall biochemical progression free survival, metastasis-free survival, and overall survival) were improved by neoadjuvant chemothermal therapy. This study has not demonstrated the any difference between high-risk but not very high-risk and very high-risk patients.

We have added new sentences “Recently, Eastham et al. demonstrated a large prospective randomized study (RP alone versus RP with NCHT) [14]. Although the primary end points, 3-year bPFS, was not met, secondary end points (overall bPFS, metastasis-free survival, and overall survival) were positive.” in the DISCUSSION section of Page 8, line 219-222.

This manuscript is a resubmission of an earlier submission. The following is a list of the peer review reports and author responses from that submission.

Round 1

Reviewer 1 Report

This review article that submitted by Sasaki T et al. entitled “Neoadjuvant chemohormonal therapy before radical prostatectomy for Japanese patients with high-risk localized prostate cancer” analyzed biochemical progression-free survival, overall survival as well as compared the efficacy of NCHT between the nVHR and VHR groups. The authors provided the detail information in patients characteristics and compared their results with several retrospective study. As discussion paragraph, the major concern for this study is the small number of patients, otherwise, this study should be more convincing for NCHT was safe and feasible.

Reviewer 2 Report

  • In this study with a substantial median FU, neoadjuvant NCHT is reportedly beneficial, in terms of biochemical recurrence-free survival, in the high-risk group, but not in the very high-risk group of patients with localised PCa.
  • The small number of patients is a major drawback of the study and hampers the extract of safe conclusions.
  • The reported difference in biochemical recurrence-free survival is an interesting, yet only statistical result. Differences in biochemical progression-free survival may be considered as clinically “meaningless”, as they do not always translate into differences in survival. From a clinical point of view, there is nothing to prompt clinicians to use NCHT in the very high risk group in order to improve survival. This was also the case with the Alliance Study, which clearly stated that NCHT therapy should not be considered as a routine clinical practice in patients with high-risk, localised PCa.
  • Overall, the study provides us with little new information regarding neoadjuvant clinical practice in high- and very high-risk PCa patients

Reviewer 3 Report

The study by Sasaki et al., analyzed the differences in responses to neoadjuvant chemo hormonal therapy (NCHT) between clinical and pathologic variables for patients. This prospective study clarified a better biochemical progression free survival rate in the high-risk, but not in not-very high risk (nVHR) group. The study suggests that NCHT is safe and feasible. However, more extensive treatment modalities are needed to improve outcomes, especially in VHR patients. Though it is not emphasized, like many other studies, these study also revealed that there is treatment associated adverse effects. The authors can mention that in the discussion and is the patients are managed differently with additional agents for adverse effects. Considering the prostate cancer survivors is highest among cancer survivors, the detailed description of the treatment and related adverse effects is crucial for weighing future treatment/management.  

Though the Asian population is not shown to be associated with the diet and obesity, the inclusion of such detail will greatly help the future researchers.

The study is clear and executed well. The study also mentioned the strength and weakness based on the small sample size.

The manuscript is clearly written without any grammatical errors/typos.